# Locomotory Profiles in Thoroughbreds: Peak Stride Length and Frequency in Training and Association with Race Outcomes

**DOI:** 10.3390/ani12233269

**Published:** 2022-11-24

**Authors:** Charlotte Schrurs, Sarah Blott, Guillaume Dubois, Emmanuelle Van Erck-Westergren, David S. Gardner

**Affiliations:** 1School of Veterinary Medicine & Science, University of Nottingham, Sutton Bonington, Loughborough LE12 5RD, UK; 2Arioneo Ltd., 94 Boulevard Auguste Blanqui, 75013 Paris, France; 3Equine Sports Medicine Practice, 83 Avenue Beau Séjour, 1410 Waterloo, Belgium

**Keywords:** horse, exercise, stride, performance, heritability

## Abstract

**Simple Summary:**

Racehorses compete in short (‘sprinters’); medium (‘milers’) or long distance (‘stayers’) races. Sprinters are thought to naturally have a shorter stride than stayers; but no study has objectively tested this theory. Here, using known race distance to categorize racehorses into one of the three aforementioned categories together with a stride tracking device that objectively measures locomotion; this study demonstrates that peak stride length in racehorses is a heritable trait that is different in sprinters versus stayers prior to them even racing at that distance. In training, sprinters took shorter strides of higher frequency and were faster to cover furlongs in race-speed training sessions from a standing start than stayers. These stride data were recorded during training sessions before the horses raced and thus categorised as ‘sprinters’ or ‘stayers’. Stride length during training did not predict later racing success. This study provides the first objective insight into locomotory differences between sprinters and stayers. Such information when coupled with the trainer’s experience/eye could help them choose the most suitable race for each individual horse; to benefit both its health and safety on the track.

**Abstract:**

Racehorses competing in short (i.e., ‘sprinters’), middle- or longer-distance (i.e., ‘stayers’) flat races are assumed to have natural variation in locomotion; sprinters having an innately shorter stride than stayers. No study has objectively tested this theory. Here, racehorses (*n* = 421) were categorised as sprinters, milers or stayers based on known race distance (*n* = 3269 races). Stride parameters (peak length and frequency) of those racehorses were collected from prior race-pace training sessions on turf (*n* = 2689; ‘jumpout’, *n* = 1013), using a locomotion monitoring device. Pedigree information for all 421 racehorses was extracted to three-generations. In training, sprinters had a shorter stride of higher frequency and covered consecutive furlongs faster than stayers (*p* < 0.001). Relatively short or longer stride did not predict race success, but stayers had greater race success than sprinters (*p* < 0.001). Peak stride length and frequency were moderately heritable (*h*^2^ = 0.15 and 0.20, respectively). In conclusion, differences in stride were apparent between sprinters and stayers (e.g., shorter stride in sprinters) during routine training, even after accounting for their pedigree. Objective data on stride characteristics could supplement other less objectively obtained parameters to benefit trainers in the appropriate selection of races for each individual racehorse.

## 1. Introduction

In flat racing, Thoroughbreds compete in various types of races usually categorised as short (<1500 m), middle (1600–2500 m) or long (>2500 m) distance. Racehorses are usually considered to be naturally predisposed to one type of race distance, due to various physiological and morphological characteristics such as size, musculature [1], and stride [2,3]. Genetics is also important [4], although the precise contribution of genetics versus environmental variables which classify successful sprinters or stayers are relatively undefined. Nevertheless, subjective information is often used by many buyers and trainers of racehorses to assign them to become predominantly short or longer-distance performers. Even so, many racehorses initially race at shorter-distance, but subsequently perform better at longer-distances (i.e., ‘sprinter-miler’ or miler-stayer’). It is a common assumption and practice that racehorses are trained similarly, regardless of their labelling as a ‘sprinter’ or ‘stayer’. In human sport, the training regimes of 100 m sprinters will contrast markedly to marathoners; short anaerobic bursts of speed requiring high muscular energy versus high aerobic capacity, efficient fuel utilization and fatigue resistance [5,6]. If trainers could complement their own assessment of a racehorses’ best distance (i.e., subjective experience or their ‘eye’) with objectively obtained training data that classified the racehorses on locomotory characteristics that distinguished a sprinter from a stayer, then more specific training sessions could be implemented to increase the chances of better performance earlier in the racehorses’ careers: horses would race at their appropriate distances and wastage could be reduced. Monitoring speed and stride length over time allows trainers to identify or anticipate musculoskeletal injuries early on during racehorse training [7].

Early determination of any type of racehorse involves complex decisions and multiple parameters. For example, shorter distance races (i.e., those at a distance of <7 furlongs or ~1400 m) require explosive speed and rapid, short strides to quickly reach maximum speed. Such a racehorse is often of shorter stature and greater muscularity, much like human sprinters. In contrast, racehorses that excel over longer-distance (>12 furlongs or ~2500 m) require stamina, often associated with leanness and longer, strides. Because such disparity in phenotype can underpin sports performance, genetic testing has grown in popularity across the racing industry [1,4]. Variation in single-nucleotide polymorphisms of the myostatin gene (MSTN), which controls muscle development, has shown that nearly all sprinters are homozygous (‘C/C’) at the MSTN locus, while heterozygous (‘C/T’) horses tend to favour middle-distance races (7–12 furlongs; 1400 m–2400 m). Racehorses homozygous (‘T/T’) appear better suited to longer distance races (>10 furlongs, 2000 m), according to previously obtained race performance [4,8,9].

It is axiomatic that in order to win any competition based on speed, the fastest individual will get to the finish line first. Racehorses increase speed firstly by increasing stride frequency (SF) up to a pace consistent with gallop (45+ kph) and then by increasing stride length (SL; 45–65+ kph) [3,10]. Therefore, in sprint races of shorter distance, the racehorse that is able to rapidly increase, or maintain a higher stride frequency, is more likely to achieve a higher speed and good performance. Over longer distances, longer stride becomes more important, alongside endurance capacity [11]. For decades, breeders and trainers have attempted to relate the physiological characteristics of racehorses during their training to their race-day performance [12]. Objective measurements of locomotory parameters have only recently become available for racehorse trainers, allowing them to potentially ascertain whether a young racehorse has a greater aptitude for sprinting or longer-distance races [3,13]. However, to date, no study has related race performance over multiple seasons with information on locomotory profile in training–do racehorses that have only raced in sprint races demonstrate an innately shorter stride early on in training? With the advent of smart devices that record multiple parameters in the equine athlete, the possibility for such an early insight into locomotory differences between sprinters, milers or stayers is now possible. A better understanding of individual horse stride characteristics could help racehorse professionals select suitable race distances, while also taking into account their own experience at placing racehorses in suitable meetings alongside other historical aspects of how racehorse conformation and pedigree information can influence such decisions. No study has specifically evaluated stride patterns in different types of racehorses (stratified by performance in short, middle or longer distance races) and retrospectively assessed locomotory profile in training in the same racehorses, considering their pedigree information, to account for the influence of genetics on racing outcomes.

Hence, in the present study, an observational study was conducted, using a fitness tracking device to study peak stride (length and frequency) in racing Thoroughbreds categorised according to the type of turf race they have participated in (sprint, mile or staying’ race; based on distance) and their subsequent racing result (win/podium/top5). Using this classification of racehorses according to their race distance, we have retrospectively classified the training data of the same horses galloping at race-speed on turf to observe whether any differences in locomotory parameters were apparent in training sessions prior to and during subsequent races. We were further able to determine whether any training parameters within each category of racehorse could predict race performance. The primary hypothesis of the study is that racehorses categorised according to race distance (sprinter/miler/stayers) may be distinguished in training by having relatively short, medium or long stride, respectively when analysed at race speed (soft-medium-hard gallop session or ‘jumpout’). Secondary hypotheses are: (1) sprinters-milers-stayers with relatively short or long stride within each category are more successful in their respective races, and (2) that locomotory parameters have moderate-to-high heritability. Finally, since many racehorses compete in different types of races (e.g., sprint, mile and/or staying race) we analysed the extent to which locomotory profiles (e.g., peak stride frequency, length) in training evolved over time within individual horses throughout the race season.

## 2. Materials and Methods

### 2.1. Databases

This retrospective, observational study used three large datasets, all including the same cohort of racehorses:(1)racehorse training sessions: collected by means of a fitness tracker (the ‘Equimetre’^™)^ from a single racing yard (Ciaron Maher Racing) in Victoria, Australia(2)racehorse pedigree information: publicly available and downloaded from https://www.pedigreequery.com (accessed 4 August 2022)(3)race results: available upon subscription in Australia, with race data recorded and downloaded from http://www.racing.com (accessed 7 July 2022)

### 2.2. Designation of Racehorses according to Race Distance

A total of *n* = 421 racehorses participating in a total of *n* = 3269 races were included in this study. Races were categorised according to class of race. In Australia, Group and Listed races are those established by the Australian Racing Board to reflect the highest standard of racing for races run in Australia. Group 1 are the highest-class races, followed by Group 2, Group 3 and Listed races. In this study, the highest-class races (*n* = 347, 10.6% of total) comprised; Group 1 (*n* = 65 of 3269 races; 1.9 %), Group 2 (*n* = 55 races, 1.6%), Group 3 (*n* = 97 races, 2.9%) and Listed (*n* = 130 races, 3.9%), whereas all other races were classed as Uncategorised (*n* = 2922 races; 89.7%). Race distance was known from http://www.racing.com (accessed on 28 October 2022) and was classified for this particular study as a ‘sprint’ race <1600 m, ‘mile race’ 1601–2500 m or ‘staying race’ >2501 m. All races were conducted on turf between 20 March 2020 to 13 May 2022. Five types of racehorse were created: (1) pure sprinter–exclusively racing over sprint distance only (*n* = 265 horses, 1563 races); (2) sprinter-miler, competing predominantly in sprint but also some mile races (*n* = 81 horses, 775 races); (3) miler, pure mile races only (*n* = 22 horses, 167 races); (4) sprinter-miler-stayer, competing in all types of race (*n* = 37 horses, 327 races) and (5) stayer-miler, purely or predominantly racing at stayer distance with some mile races (*n* = 16 horses, 131 races). Racehorses were aged between 2–10 years of age at the time of racing and included males (colts/stallions; *n*= 33 average age, 2.93 ± 0.42 years), females (fillies/mares; *n*= 197, 3.88 ± 0.90 years), geldings (*n* = 174, 4.67 ± 1.42 years) or of unknown/unrecorded sex (*n* = 17, 3.76 ± 0.69 years). All race data were extracted online from http://www.racing.com, accessed on 28 October 2022. Other aspects of the dataset such as venue, track condition, carried weight, handicap, rating and prize money were recorded.

### 2.3. Training Data

Horses wore their regular tack and were exercised by a randomly allocated work rider, who varied according to individual training sessions. A tracking device (‘Equimetre™’, Arioneo, Ltd. Paris, France) was fitted to the girth prior to training by persons accustomed to using the device, as previously described [14]. The device recorded locomotory parameters (peak stride length and frequency) alongside speed (by GNSS) and cardiovascular parameters (peak HR), as previously described in detail [3]. The trainer determined the nature of each individual training session, directing the work-rider as appropriate. The Equimetre was not systematically placed on each horse for every individual training session, rather for specific sessions. From the GNSS (GPS + Glonass + Galileo) satellite data, speed (i.e., time taken to cover 200 m in seconds) recorded for each 200 m segment (at 200, 400, 600, 800, 1000, 1200 and 1400 m) was recorded. The fastest 200 m was then used to designate the session as soft, medium or hard gallop. All training sessions at gallop were conducted on turf. In addition, a separate dataset of ‘jumpout’ training sessions were available for analysis with similar logged data. These sessions aim to replicate race-day barrier trials and conditions. Horses of similar ability are grouped to ‘race’ simultaneously from starting gates for the duration of the training session. All jumpout sessions were also conducted on turf. All training data were collected between 7 April 2020 to 19 April 2022 and comprised a total of 2689 training sessions, with 12 (8–19) median (first-third interquartile range [IQR] per racehorse. From the exact date of training, together with the exact race date, the number of days prior to each race plus the interval in days between races could also be recorded for each horse. Final datasets were checked for artifacts and corrected accordingly in MS Excel. Environmental temperature and precipitation were recorded as potential covariates in any analyses. Using the hard outcome of race performance in races of known distance to classify five categories of racehorse from sprinter to stayer, then the same categorisation was applied retrospectively to all 421 racehorses during their gallop training sessions that occurred prior to, and during the two race seasons as recorded here (2020 to 2022).

### 2.4. Pedigree Data

For each individual racehorse, an online search was first conducted on the Thoroughbred Pedigree Database http://www.pedigreequery.com, accessed on 28 October 2022 to obtain a three generation pedigree for all 421 individual racehorses The resulting pedigree dataset consisted of *n* = 2690 horses from 629 sires (259 of which were founders), and 1628 dams (693 founders). Where racehorses were either not present or multiple racehorses with the same name existed, then data were cross-checked using a further database (Equineline.com). The data for each individual racehorse was then manually reverified on http://www.racing.com/horses/, accessed on 28 October 2022 for trainer, horse age, sex and racing profile. As a further check, using a random number generator in MS Excel (between 001–421), ten further racehorses were cross-checked for accuracy. The final three generation pedigree was used to estimate heritability.

### 2.5. Statistical Analysis

Any normally distributed descriptive data (e.g., peak stride length, stride frequency) are presented as mean (±1 standard deviation [SD]). Similar data that were not normally distributed or categorical are presented as median (1st–3rd interquartile range) or as percentage (of total number) for categorical variables. Data distribution was checked either by standard tests (e.g., Shapiro–Wilk test) or checking of residuals post analysis. If necessary, data was log-transformed (log_10_) to normalise the distribution of the data prior to analysis. For some analyses, where assumptions for analysis of variance (ANOVA) could not be met due to occasional missing data (e.g., artefacts removed or no data present), linear mixed models (restricted maximum likelihood; REML) were used with the main effect of interest fitted as a fixed effect and HorseID or racecourse fitted as random effects. This statistical model assumes that occasional missing data are distributed at random amongst fixed effects. Other potentially confounding factors that were not part of the design but may influence outcome, as assessed by univariate analysis (*p* < 0.10) were included as co-variates (e.g., interval between race days, temperature, precipitation). Estimates of heritability were generated using a sparse inverse relationship matrix (’ainv’) generated for all 421 racehorses and their three-generation pedigree. Combining the pedigree file with phenotypic outcomes such as stride length in an animal model (i.e., using REML) allowed us to obtain variance parameters and narrow sense heritability estimates (additive genetic variance; Genstat v21, VSNi, Rothampsted, Harpenden, UK). Approximate standardised error (SE) for *h*^2^ was obtained using the delta method, which uses a Taylor’s expansion to get the variance of a function of a parameter (Var(f(x)) = Var(x) × (f’(x))^2^). Estimates were obtained after adjusting for age and including the type of racehorse (e.g., sprinter versus stayer) as fixed effects. Significant variation in the proportion of wins/placing according to the type of racehorse (‘sprinter versus stayer’) was analysed by logistic regression fitting win, top3 or top5 as individual binomial outcomes (yes/no) and type of horse as a fixed effect adjusted for any significant confounding variables (age of horse, track condition, race class). All data were analysed using Genstat v22 (VSNi Ltd., Rothampsted, Harpenden, UK). Statistical significance was accepted at *p* < 0.05.

## 3. Results

### 3.1. Racing Data and Performance

Within each category, racehorses participated in a similar number of races: (pure sprinter, 4 [1,2,3,4,5,6,7,8] races); sprinter-miler, 7 [1,2,3,4,5,6,7,8,9,10,11,12,13,14] races; miler, 6 [1,2,3,4,5,6,7,8,9,10,11] races; sprinter-miler-stayer, 8 [1,2,3,4,5,6,7,8,9,10,11,12] races; stayer, 7 [2,3,4,5,6,7,8,9,10,11,12] races, median [1st–3rd IQR]. Sprinters were significantly younger than milers, who were younger than stayers at the time of racing: (pure sprinter, 3.6 ± 0.9 years; sprinter-miler, 4.6 ± 1.2 years; miler, 4.8 ± 1.3 years; sprinter-miler-stayer, 4.9 ± 1.0 years; stayer, 6.6 ± 1.3 years mean ± 1 SD). Average prize money won according to the class of the race was significantly different between race classes, (Group 1, $93,898 ± 85,880; Group 2, $38,955 ± 42,618; Group 3, $28,573 ± 29,159; Listed, $27,012 ± 33,646; Uncategorised, $9847 ± 12,938). For *n* = 35 of 3268 races the final position was unknown. Overall, 508 races were won by 255 different racehorses, 331 different racehorses achieved a top three placing in a total of 2175 races and 375 of 421 racehorses were placed top five in a total of 1668 races. The remainder were unplaced. Racehorses therefore either won or were placed top three or top five in 15.5, 36.4 or 51.0% of races, which varied significantly according to the type of horse (Table 1). ‘Stayers’ were less common, competed in fewer races but were more successful than sprinters (Table 1). Race distance (meters) was not different (*p* > 0.05) between different class of race (Group 1, 1712 ± 664; Group 2, 1624 ± 427; Group 3, 1564 ± 501; Listed, 1762 ± 653; Uncategorised, 1566 ± 571 m).

### 3.2. Training Data and Locomotory Performance

All sessions were effectively ‘race-pace’, as illustrated in Table 2, with horses covering a furlong (200 m) in 10–12 s, achieving speeds of up to 67 kph, with peak stride frequencies and length increasing with speed and indicative of race-pace efforts, as previously described. However, when categorised according to the type of racehorse, then significant differences were apparent; sprinters *per se* had significantly shorter peak stride length and higher frequency than stayers, with a gradual change between intermediary categories (Table 3). The expected increments in peak stride length and frequency with harder training sessions were observed across all categories of racehorse (Table 3).

In further data, for 378 of the 421 horses where speed and locomotory information were available for pre-race ‘jumpout’ training sessions (*n* = 1013); that is, starting from a standing start in stalls, then the best time to cover any 200 m (furlong) from the first to the fifth furlong at 1000 m, then speed gradually became slower for all categories of horse, as expected, but was always significantly faster for retrospectively designated sprinters versus stayers (Figure 1a). In addition, peak stride frequency was higher (Figure 1b), stride length was shorter (Figure 1c) and peak recorded speed was higher (Figure 1d) in sprinters versus stayers. During the course of two racing seasons, we described whether locomotory parameters changed within individual racehorses of each category (sprinters to stayers; Figure 2). Whilst significant variability with individual training session from the first to tenth (relatively few horses completed ≥10, data censured at *n* = 10 training sessions) existed for peak stride length (Figure 2a) and peak speed (a slight increment; Figure 2c), the effect sizes were relatively small and not consistent, suggesting little evolution of locomotory parameters (i.e., no effect for peak stride frequency, Figure 2b,d).

### 3.3. Training Data, Type of Racehorse and Predicting Race Outcome

Combining locomotory data during training for each individual racehorse with their race outcomes (win or top three ‘podium’) suggested that colts were more likely to win than geldings, with the chance of winning a group race declining as the class of race increased (Figure 3a). There was a trend for racehorses with longer peak stride length (i.e., stayers) to have an increased chance of winning and finishing in the top 3, regardless of race class and racehorse type (Figure 3b).

### 3.4. Heritability of Stride Parameters and Peak Heart Rate

After accounting for pedigree, which incorporates all traits with high genetic potential that were not recorded in our dataset (e.g., height, musculature etc…) then the difference in stride parameters recorded during training, according to the type of racehorse, was maintained (e.g., sprinters having shorter stride with higher frequency than stayers; Table 4).The estimates of narrow-sense heritability were significant (as determined by change in the log^2^ deviance ratio when pedigree information was included or not) and the values of *h*^2^ were low to moderate (Table 4). The heritability of peak heart rate was however not significantly different across racehorse categories (Table 4).

## 4. Discussion

This study has directly, and objectively, outlined how differences in locomotory profile (short, medium or long peak stride length/frequency) are already apparent in sprinter versus stayer racehorses, during race-speed training sessions. Indeed, during mock-races from a standing start, i.e., ‘jumpout’ sessions, sprinters also achieved higher speeds than stayers. Nevertheless, racehorses with relatively short or long stride within sprint or stayer categories, respectively did not predict race performance (i.e., winning or podium position). However, we are able to report for the first time that locomotory parameters of racehorses have moderate heritability. Therefore, the study provides evidence to support our primary hypothesis, that racehorses can be distinguished in training by having relatively short, medium or long stride. Whilst locomotory parameters have moderate heritability, locomotory parameters in individual racehorses do not appear to alter considerably during the course of a race season. Hence, we suggest that differences in locomotory profiles are tangible for each type of racehorse (e.g. sprinter, miler, stayer), necessitating unique stride and speed aptitudes for the required distance. Race-day performance remains complex and may be influenced by a multitude of factors including sex of the horse, stride and to some extent pedigree as described in this study.

### 4.1. Racing Data and Performance

In Australia, horses race all year long with the season running from August to July, including a period of ‘spell or detraining’. A spell or period of detraining refers to an extended period, usually 6 to 8 weeks, during which a racehorse is given a rest in the paddock. This break is dependent on the number of race starts completed during that year [15]. Thoroughbreds begin racing at the age of two and often progress from relatively shorter to longer distances (~above 1600 m) as their stamina and musculature develops, usually when they reach the age of three [16]. Previous research has shown that 2 year old racehorses are more suited to shorter races than any other age group [4,17,18]. Older racehorses, between 4 to 5 years of age, are therefore more likely to race over longer distances (i.e., 1600–3200 m) [19]. These trends were consistent with the ages observed across the three racehorse profiles (sprinter, miler or stayer) in this study; younger horses tend to be sprinters, whilst milers and stayers were significantly older.

Racehorses being trained in Australia are predominantly sprinters. 39% of Group races in Australia are run over less than 1400 m compared with 23% in both the UK and Ireland [20,21,22]. This may be partially explained by the fact that, in some racing nations, a premium is allocated to horses participating in shorter distance races, as the prizemoney/class of the races tends to be higher than other race distances. Sprinters largely characterised our dataset; we observed a much higher proportion of sprinters with both race and training data than stayers (according to a race distance classification established by [23]). This is not surprising considering the strong selection for early speed which characterizes the Australian racing industry. However, there has been a surge of global initiatives to boost and encourage the breeding of stayers to counteract this phenomenon [24]. Evoking greater prestige and higher prize money, some examples of long-standing stayers’ race include The Epsom Derby (UK; 2420 m), Prix de L’Arc de Triomphe (France; 2400 m), Breeders’ Cup Classic (USA; 2000 m) and the Melbourne Cup (Australia; 3200 m).

The limited number of pure stayers in our dataset nevertheless presented the highest proportion of race wins, compared to other categories of racehorse. Stayers were significantly older, were possibly more mature and of better ‘quality’ and thus retained to race or were better placed in suitable races given greater knowledge about their optimal characteristics. Younger, less talented and successful horses may also have dropped out of the yard and thus dataset. To an extent, therefore, perhaps such longer distance races are comprised of more appropriately placed and better racehorses. Additionally, in longer distance races, there is more opportunity for jockeys or trainers to utilize racing tactics [25]. Interestingly, a racehorse’s peak racing age was previously suggested to be 4.45 years [26], a two-year difference with the stayers in this study.

### 4.2. Training Data and Locomotory Performance

Racehorse athletic careers generally only span a few years, during which racing opportunities can be limited [15]. Opting for a race distance that matches the individual horse’s characteristics and racing ability could markedly contribute to increasing its chance of winning. Therefore, trainers subjectively determine individual racehorse locomotory profile (sprinter, miler or stayer) early on in their training in order to ideally target the most appropriate exercise program and maximize their racing performance. Yet, a racehorse’s ability to gallop over five furlongs for a sprint race, as opposed to twenty for a stayer’s race, will differ significantly in terms of locomotion strategy. As they approach peak speeds, individual horses will either naturally increase their peak stride length or frequency. Over shorter distances, the requirements for acceleration and speed are pivotal, but as the distance increases, then efficiency of stride and stamina become more important. Stride length, rather than frequency is the main determining parameter to achieve higher maximal speeds [3]. During standard gallop training sessions, our results revealed clear locomotory differences: sprinters had shorter stride length of a higher frequency than stayers. It is conceivable that the effect of warm up, if different between sprinters and stayers (not to our knowledge) may have exerted some effect on these stride characteristics, as previously evidenced in showjumpers [27]. Fatigue can also cause racehorses to lose a stable stride frequency [28], triggering a decrease in stride length [29]. We were unable to account for such aspects in our study, but in-field biomarking of fatigue through spot-sampling of blood could reveal important differences that could be trained in.

In preparation for racing, racehorses are often exposed to ‘jump-out’ training sessions, which consist of grouping horses of similar age/level, to start from barrier stalls and to race against each other under timed conditions. This race-day simulation exercise is different to official race-day barrier trials [30], but, from the horses’ perspective, is akin to a race. Unsurprisingly, speeds recorded during such sessions were among the highest recorded in our dataset yet were still significantly higher for sprinters compared to milers/stayers. Thus, in both regular training sessions and race-speed simulation sessions, sprinters and stayers could be clearly differentiated on stride characteristics. Nevertheless, we did not note any significant evolution of stride characteristics through incremental training sessions over the course of two race seasons (~10 training sessions per horse). Such differences are likely small for any individual horse and it is likely that a very large dataset would be required to observe significant differences to validate the evolution of stride over a racehorse’s career. Previous research has suggested that a typical racehorse improves its race time by approximately 10 (horse) lengths in sprints of <1 mile and up to 15 lengths for middle-longer distance races (≥1 mile) from the age of 2 to 4.5 years [26]. Hypothetically speaking, a proportion of this improvement could be attributed to alterations in the speed or efficiency of locomotion, although this was not measured in that study.

### 4.3. Training Data, Type of Racehorse and Predicting Race Outcome

Racehorse success on the track results from a complex combination of genetics [31], nutrition [32] and training [33]. Such factors determine the expression of physical traits specific to the athletic demands of the sport. In Thoroughbreds, muscle strength, speed and endurance have been identified as traits that favour superior performance at various race distances [34]. In this study, colts (i.e., younger male horses not gelded) had higher odds of winning than geldings. This aligns with previous research which identified some of the non-genetic factors that affect racehorse performance: sex, age, class of race, track condition, handicap weight and distance [35], and suggests that younger ‘entire’ male horses have the greatest chance of winning a race. Such an observation is also at odds with the fact that, in our dataset, the few stayers, who were older, had more race success. Perhaps the greater competition between horses in sprint races, the predominant race category in our dataset, is primarily won by colts as opposed to geldings. In stayer races, primarily competed in by older mares and geldings, and for obvious reasons very few entire males, then such differences are not apparent.

Thoroughbreds present unique musculoskeletal characteristics compared to other breeds. Notably, they have a large mass of skeletal muscle, low body-fat proportion and a greater percentage of fast twitch muscle fibres [36]. The composition of muscle fibre type, namely in the propulsive gluteal muscles, evolves with age and training, progressively improving stamina [37]. Previous work on racehorse wither height, has also revealed interesting insights. In mature horses, wither height was positively correlated with racing performance [38] and stride length [39]. The relationship between conformation and stride variables in foals aged 6–8 months has also been studied: increased speed was attained by longer stride length in heavier foals and higher stride frequency in taller foals [40]. The effect of training on performance has also been examined. [37] outlined that training strategies targeting both strength and endurance concurrently impinge on performance when compared to training programmes aimed at optimizing either one or the other [37]. This is explained by the fact that strength for acceleration is required for a sprint race. Strength is associated with an increased muscle mass, a shift from slow twitch to fast twitch muscle fibres and an increase in ATP utilisation. As a result, adaptations for sprinters would be disadvantageous for stayers as they rely on slow twitch muscle fibres and aerobic metabolism.

In Standardbreds and Thoroughbreds, after three years of training, changes in the trotting strides were observed: stride length, stride duration and swing phase increased [41]. Training plays an important role in the development of the above parameters [42]. Our findings highlighted a tendency of increased odds of winning and/or finishing in the top 3 in horses displaying a longer stride length. Similarly, in harness trotters, a test performed on the track showed that performing horses presented the highest maximal stride frequency and a long stride length [43].

### 4.4. Heritability of Stride Parameters and Peak Heart Rate

For the last three centuries, Thoroughbreds have been intensely bred for their elite athleticism, stamina and aptitude for speed. Racehorse pedigree information is registered in The General Studbook, 1791 [44] and can be traced back to their original ancestry. The Thoroughbred genetic pool is narrow and emerges from three foundation stallions (Arab, Barb and Turk) and approximately 30 foundation mares (UK) [45,46,47]. Since then, the continuity of the breed has become ever more controlled and refined. For example, English and Irish breeding industries focused on producing distinctive types of horses from precocious, fast, 2-year-old sprinters, ‘classic’ middle-distance runners or horses with enhanced stamina suitable for less popular ‘classics’ races such as the St Leger (United Kingdom; 2900 meters). Heritability of any given trait refers to the percentage of the parental trait that could effectively be transmitted to its offspring. Calculations on narrow-sense heritability estimates (*h*^2^) for specific traits such as stride length aim to estimate the strength of genetic determinants for the particular characteristic (human height being highly heritable at 0.85), with the remainder being non-genetic, additive, environmental effects such as trainer, rider, track, etc. In flat racing, heritability of performance has been estimated as relatively low with wide confidence intervals, e.g., *h*^2^ between 0.15 to 0.55 [35,48]. The heritability of locomotory characteristics (speed, stride length and frequency) has been estimated previously in French saddle horses and was considered to increase with pace (e.g., from walk [*h*^2^ = 0.23], trot, canter through to gallop [*h*^2^ = 0.52]) [49]. In this study of racing thoroughbreds, heritability of peak stride length and frequency was moderate (*h*^2^ = 0.15–0.20), suggesting that 15–20% of the variation in locomotion in Thoroughbreds is due to the particular genes each racehorse inherited, with the remaining 80–85% of the variation due to environmental (non-genetic) factors. Since peak heart rate was consistently measured in our training data, we thought it interesting to also assess its heritability, despite previous papers indicating that between-individual variability is likely high [3,50,51]. Nevertheless, similar to locomotory characteristics heritability of peak HR was moderate (*h*^2^ = 0.19).

Finally, it should be recognized that the study is a convenience sample of racehorses having used an ‘Equimetre’ intermittently, but in a repeatable fashion in training. The study is retrospective and observational. One test of our data would be to prospectively assign racehorses based on stride characteristics in training to being a sprinter or stayer and observe whether greater success in that category was achieved. Other factors that could influence stride but were not recorded consistently such as different warm-up protocols or accumulated fatigue were not taken into account. Since many racehorses compete in different types of race (e.g., sprinter or mile, mile or staying race) it would be interesting to analyse the extent to which locomotion (e.g., peak stride frequency, length) evolves over the course of a racehorse’s career for different profiles (sprinter, miler, stayer).

## 5. Conclusions

In conclusion, this study demonstrates that locomotory differences exist (peak stride length, frequency) between various types of racehorse (sprinters, milers and stayers). Stride characteristics measured at the onset of training can predict aptitude to racing in a given category, regardless of the potential progress obtained with training. Peak stride length is a moderately heritable trait that can be bred for. Considering heritability of stride along with objective locomotory data and other aspects (i.e., preferred ground, going), may also help trainers choose early on what type of training (short, middle- or long-distance work outs) or which race to enter (distance, profile, going etc…). Such a hybrid approach using data alongside experience may contribute to improved welfare on the track and prolong racing careers.

## Figures and Tables

**Figure 1 animals-12-03269-f001:**
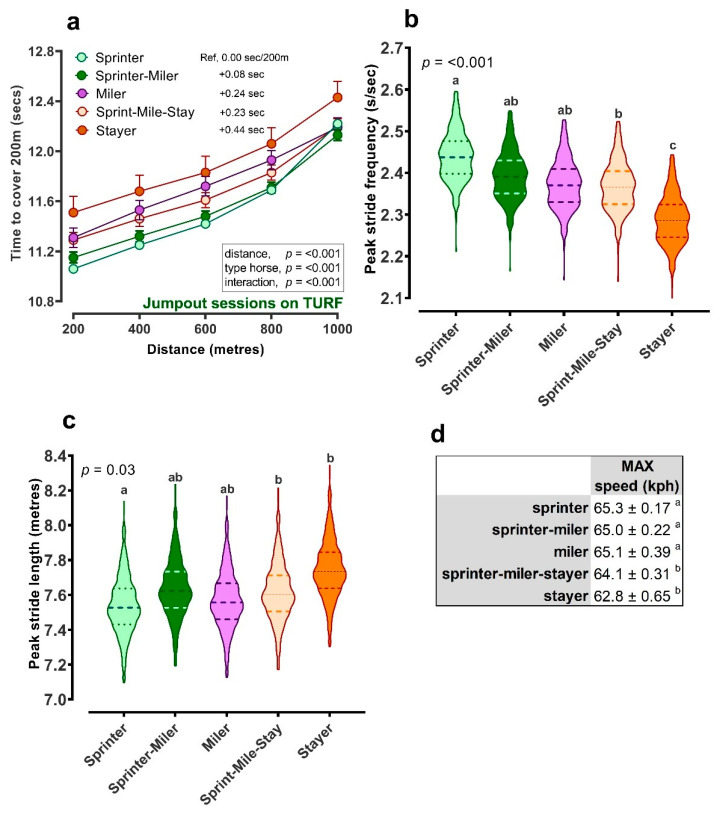
Training data during jumpout for different types of racehorse. (**a**) data are estimated marginal means ± SEM for consecutive 200 m segments (**b**) data are for all horses’ peak stride frequency or (**c**) peak stride length during jumpout training sessions, (**d**) mean ± 1SD speed, according to category of racehorse. All data obtained from an ‘Equimetre’ used at a single racing yard in Victoria, Australia (*n* = 378 different racehorses, *n* = 1013 different ‘jumpout’ training sessions). Type of racehorse applied retrospectively to training data after competing in races of known distance. Data analysed by Restricted Maximal Likelihood (REML), as described previously (e.g., see Table 3). Differing superscripts are significantly different at *p* < 0.05.

**Figure 2 animals-12-03269-f002:**
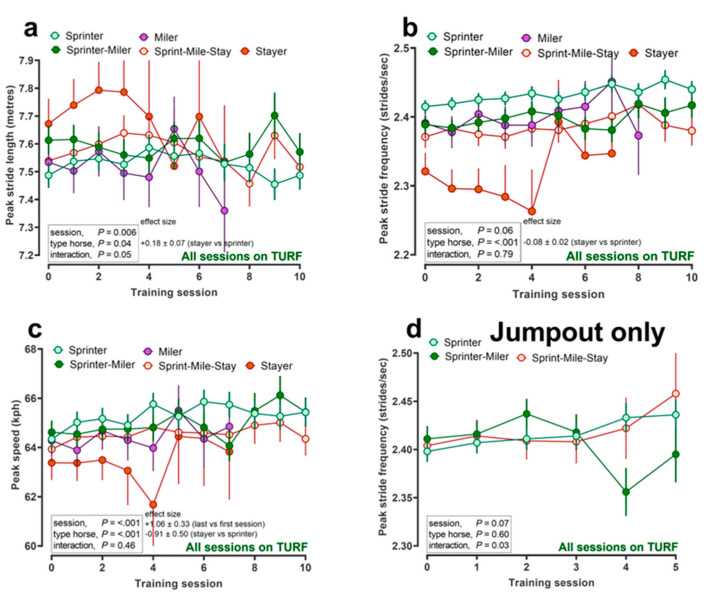
Evolution of locomotory parameters throughout the training season. Change in locomotory parameter with number of training sessions (‘session’) for each type of racehorse (‘type horse’). Racehorses were categorised based on competing in races of known distance, as described in Methods. (**a**–**d**) data are estimated marginal means ± SEM for consecutive training sessions (x-axis) over the course of two racing seasons, adjusting for age of racehorse and interval between training sessions. Individual racehorse, year/training month were included as random effects. Data were obtained using an appropriately fitted ‘Equimetre’ device from a single racing yard in Victoria, Australia.

**Figure 3 animals-12-03269-f003:**
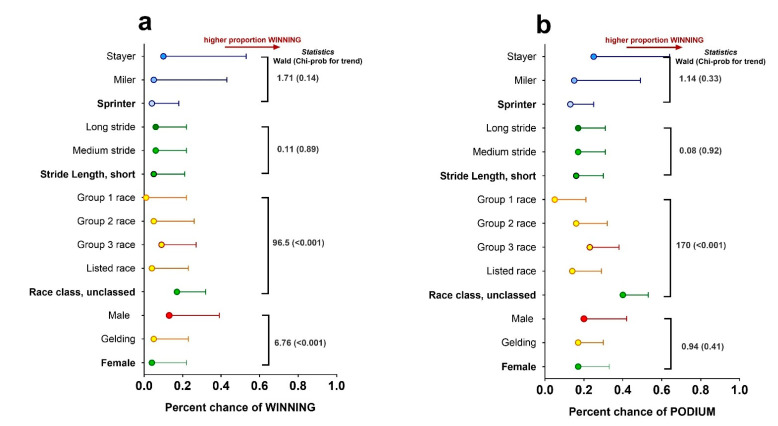
Predicting race wins or podium places on training data. Predicting race wins or podium places on training data. Data are predicted mean ± SEM for (**a**) winning a race (left graph) or (**b**) achieving a top three placing (right graph). Predicted means were obtained by integrating known race outcome data from Racing.com (Victoria, Australia; *n* = 421 different racehorses, *n* = 3269 different races) with training data (only gallop on turf) for the same horses before and during two race seasons (*n* = 1799 different training sessions). Data were analysed with race outcome (e.g., win or top three) as the variable of interest fitting sex of horse, race class, stride length and age as fixed effects in the model. Since the same horse completed multiple training sessions and multiple races then the individual horse was fitted as a random effect using Generalised Linear Mixed Models (GLMM). Statistics for main effects are indicated on the right on the graph with associated Wald statistic and χ^2^ probability. Statistical significance was accepted at *p* < 0.05.

**Table 1 animals-12-03269-t001:** Race performance stratified by type of horse.

	All (*n* = 3269)	Sprinter (*n* = 1671)	Sprinter-Miler (*n* = 874)	Miler (*n* = 167)	Sprinter-Miler-Stayer (*n* = 408)	Stayer (*n* = 149)	* *p*-Value
**Wins** (%)	508 (15.5)	234 (14.0)	134 (15.3)	27 (16.2)	74 (18.1)	39 (26.2)	<0.001
**Top3** (%)	1190 (36.4)	593 (35.5)	299 (34.2)	60 (35.9)	166 (40.7)	72 (48.3)	<0.001
**Top5** (%)	1668 (51.0)	823 (49.3)	422 (48.3)	89 (53.3)	239 (58.6)	95 (63.8)	<0.001

Values are number meeting criteria for each row (proportion [%] of total races in each column). Data as per Racing.com racing records in Victoria, Australia (*n* = 421 different racehorses, *n* = 3269 different races). * analysed by logistic regression (see Section 2.5 in Methods).

**Table 2 animals-12-03269-t002:** Descriptive characteristics of the training dataset.

	Soft Gallop (*n* = 635)	Medium Gallop (*n* = 579)	Hard Gallop (*n* = 585)	*p*-Value
Best 0–200 m (s)	11.6 ± 0.1	11.3 ± 0.1	10.8 ± 0.4	<0.001
Max speed (kph)	62.5 ± 0.9	64.2 ± 0.8	66.8 ± 1.4	<0.001
Peak stride frequency (stride/s)	2.37 ± 0.08	2.40 ± 0.00	2.45 ± 0.08	<0.001
Peak stride length (ms)	7.35 ± 0.24	7.48 ± 0.24	7.63 ± 0.27	<0.001

Values are Mean ± 1SD for continuous data recorded by ‘Equimetre’ in Australia (*n* = 421 different racehorses, *n* = 1799 different training sessions). Data were available throughout the year. Training intensity (soft/med/hard gallop) was calculated from the cohort based upon the fastest furlong (200 m interval) for each session and slower intensities (slow/med/hard canter) were excluded. Such data was restricted to training sessions conducted on turf track surfaces. Data were analysed by one-way ANOVA, blocking for the individual horse to account for multiple training sessions conducted by the same horse.

**Table 3 animals-12-03269-t003:** Peak stride frequency and length in race-speed training efforts categorised by type of racehorse.

	Sprinter	Sprinter-Miler	Miler	Sprinter-Miler-Stayer	Stayer	* *p*-Value
Training distance (m)	4924 ± 1146 ^a^	4987 ± 1232 ^ab^	4993 ± 1192 ^ab^	5167 ± 1240 ^b^	5333 ± 1148 ^ab^	0.018
‘Work’ distance (m)	1802 ± 380 ^a^	1912 ± 479 ^b^	1968 ± 479 ^b^	2124 ± 565 ^c^	2292 ± 650 ^c^	<0.001
Peak stride frequency (strides/sec)						
Soft Gallop	2.39 ± 0.07 ^a^	2.36 ± 0.07 ^b^	2.38 ± 0.08 ^bc^	2.35 ± 0.08 ^bc^	2.28 ± 0.06 ^c^	Type horse, <0.001 Train intensity, <0.001 Interaction, 0.03
Medium Gallop	2.41 ± 0.07	2.39 ± 0.07	2.38 ± 0.08	2.37 ± 0.06	2.29 ± 0.08	
Hard Gallop	2.46 ± 0.08 *	2.43 ± 0.08 *	2.46 ± 0.05 *	2.41 ± 0.06 *	2.34 ± 0.05 *	
Peak stride length (meters)						
Soft Gallop	7.31 ± 0.23 ^a^	7.41 ± 0.23 ^ab^	7.37 ± 0.21 ^b^	7.41 ± 0.24 ^bc^	7.60 ± 0.20 ^c^	Type horse, <0.001 Train intensity, <0.001 Interaction, 0.07
Medium Gallop	7.44 ± 0.24	7.50 ± 0.22	7.52 ± 0.30	7.56 ± 0.22	7.77 ± 0.24	
Hard Gallop	7.61 ± 0.28 *	7.67 ± 0.27 *	7.54 ± 0.20 *	7.67 ± 0.24 *	7.88 ± 0.08 *	

Values are Mean ± 1SD for continuous data recorded by ‘Equimetre’ in Australia (*n* = 421 different racehorses, *n* = 1799 different training sessions). Horse profile (sprinter, miler, stayer) was determined based on race distance and subcategories (pure sprinter/miler, sprinter-miler, sprinter-miler-stayer) were formed according to the nature/proportion of the races for every individual horse (see Methods). Data were available throughout the period before (not more than three months) and during racing. Data were analysed by restricted maximal likelihood (REML) for the main effect of type of racehorse, with each individual racehorse included as a random effect, adjusting for significant covariates (age and weight of the horse). ^a,b,c^ Values within a row with differing superscripts are significantly different at *p* < 0.05, with Bonferroni correction for multiple testing. * Significant effect of training intensity (hard versus soft gallop).

**Table 4 animals-12-03269-t004:** Heritability (*h*^2^) of locomotory and peak heart rate during race-speed training efforts categorised by type of racehorse.

	Sprinter	Sprinter-Miler	Miler	Sprinter-Miler-Stayer	Stayer	Heritability	** p*-Value
Peak stride frequency (stride/s)	2.43 ± 0.01 ^a^	2.39 ± 0.01 ^b^	2.38 ± 0.01 ^b^	2.37 ± 0.01 ^b^	2.31 ± 0.02 ^c^	0.20 ± 0.15	<0.001
Peak stride length (meters)	7.45 ± 0.02 ^a^	7.54 ± 0.03 ^ab^	7.49± 0.05 ^ab^	7.53 ± 0.04 ^b^	7.62 ± 0.06 ^ab^	0.18 ± 0.14	0.003
Peak heart rate (beats/min)	217 ± 0.6 ^a^	216 ± 0.8 ^a^	214 ± 1.6 ^a^	215 ± 1.2 ^a^	215 ± 1.8 ^a^	0.19 ± 0.15	0.08

Values are Mean ± 1SE for continuous data recorded by ‘Equimetre’ in Australia (*n* = 421 different racehorses with three-generation pedigree). Horse profile (sprinter, miler, stayer) was determined based on race distance and subcategories (pure sprinter/miler, sprinter-miler, sprinter-miler-stayer) were formed according to the nature/proportion of the races for every individual horse (see Methods). Data are averaged values for each horse from all available gallop sessions (*n* = 1699). Data were analysed and heritability estimated by REML analysis of an Animal Model. SE for Heritability (*h*^2^) was calculated using the delta method. Values within a row with differing superscripts are significantly different at *p* < 0.05, with Bonferroni correction for multiple testing. * Overall *p*-value for comparison between groups of main effect.

## Data Availability

All data were collected by Arioneo Ltd. The race results and pedigree information are publicly available at http://www.racing.com, accessed on 28 October 2022 and http://www.pedigreeenquiry.com, accessed on 28 October 2022, whilst the training data were previously collected as part of routine recording by the racing yard, who shared the data with the external company (Arioneo Ltd.) that manufactures the data logging device (the ‘Equimetre’). Anonymised training data available at http://www.arioneo.com, accessed on 28 October 2022.

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
