# Peer review of "Locomotory Profiles in Thoroughbreds: Peak Stride Length and Frequency in Training and Association with Race Outcomes"

_animals, 2022, doi:10.3390/ani12233269_

Round 1

Reviewer 1 Report

Comments to the Author

Review: “Heritability of locomotory profiles in Thoroughbreds: peak  stride length and frequency in training and association with  race outcomes

This study is an interesting account of the effect peak stride length and frequency in training and its association with race outcomes. In addition, this study estimates the heritability of these parameters. However, there are some fundamental issues and major limitations with the work, although perhaps as the methods are not well described, so with clarification these issues/limitations may not be present.

Two mains concerns regard

1)  The material and methods is not sufficiently explained and we can find it, too often,  in the results section

2) Please be consistent (and more concise) with this terminology throughout. Avoid to repeat the same information.

I consider that this is an interesting work that can attract this journal’s readers and, therefore, can be publishable provided that some aspects are clarified and some changes are made in order to make it more understandable.

Specific comments
Introduction

L51. “one-size-fits-all” should be better explained

L52. Heresy I would choose another term and look for bibliography to support my statement.

L65 ”rangy” should be better explained

Material and methods

L118-122 I would create a specific section to describe the different databases in detail

L126-128 It should be better explained how the quality of the race was categorized, the meaning of “listed” and why almost 90% do not belong to any category.

L129-130 Why do the distances vary from the introduction?

A specific section (out of pedigree section) should be created where the estimation of the genetic parameters is better explained, the correction factors used in each parameter and why this methodology has been chosen (instead of the usual blupf90, vce or tm). Consider including the genetic correlations.

All statistical methodology developed in results (below each table or figure) must be thoroughly and neatly explained in a specific section within material and methods and eliminated from results

Results

L213 n = 35 of 3268 Delete “n=”

L218-221 The number of participants in the races is constant? This should be detailed in material and methods

Table 2 and 3 delete “Training type… “and “type horse…”

Explain better ‘spell or detraining’

Explain ‘jump-out’ t before

The section Training data, category of racehorse and predicting race outcome is confusing. Really the bibliography discussed is difficult to spin with your result. I don't know if that section should be improved or deleted.

Colts age range should be explained in material and methods

Author Response

This study is an interesting account of the effect peak stride length and frequency in training and its association with race outcomes. In addition, this study estimates the heritability of these parameters. However, there are some fundamental issues and major limitations with the work, although perhaps as the methods are not well described, so with clarification these issues/limitations may not be present.

Two mains concerns regard

1)  The material and methods is not sufficiently explained and we can find it, too often, in the results section

2) Please be consistent (and more concise) with this terminology throughout. Avoid to repeat the same information.

I consider that this is an interesting work that can attract this journal’s readers and, therefore, can be publishable provided that some aspects are clarified and some changes are made in order to make it more understandable.

We thank the anonymous reviewer for the positive comments and have, where we agree, made changes to the manuscript as detailed below.

Introduction

L51. “one-size-fits-all” should be better explained.

Phrase removed.

L52. ”Heresy” I would choose another term and look for bibliography to support my statement.

Removed and changed sentence to “the training regimes of 100m sprinters will contrast markedly to marathoners; short anaerobic bursts of speed requiring high muscular energy versus high aerobic capacity, efficient fuel utilization and fatigue resistance [5,6].” Lines 51-53

L65 ”rangy” should be better explained

Removed as already described (longer strides). L68

Material and methods

L118-122 I would create a specific section to describe the different databases in detail.

Section added, with further details. (Lines 120-128)

L126-128 It should be better explained how the quality of the race was categorized, the meaning of “listed” and why almost 90% do not belong to any category.

Changed the choice of wording: ‘race quality’ altered to ‘race class’. The class of race from Group 1 to Group 3, listed and unclassed is pretty standard terminology throughout the equine world, particularly thoroughbreds. Anyone sufficiently interested to read this manuscript will understand the terminology.

Meaning of listed added.

Why do almost 90% not belong to any category? This is because a vast majority of the cohort are ‘young’ horses, that have not yet achieved sufficiently to race in the higher class races. (lines 132-135)

L129-130 Why do the distances vary from the introduction?

We have chosen our own categorization, similar to the one stated by the Australian racing authority Racing Victoria (https://www.racingvictoria.co.au/) in the state in which all the races were conducted. Different countries have slightly different categories for sprint vs long distance races, but only varying by a few hundred metres, which would not materially effect the results in this manuscript since the differences of interest are between the two most different types of racehorse, sprinter vs stayer. 

A specific section (out of pedigree section) should be created where the estimation of the genetic parameters is better explained, the correction factors used in each parameter and why this methodology has been chosen (instead of the usual blupf90, vce or tm). Consider including the genetic correlations.

We reconsidered the title, in part because of this reviewer suggestion. The main purpose of the paper is to describe the relationship between stride length/rate and type of racehorse e.g. ‘sprinter vs. Stayer’, as opposed to the heritability aspect. However, taking genetic parameters into account whilst retaining  the effects of differences in stride rate/length between sprinters and stayers is of interest, as suggests the differences are not due to genetically driven differences (size, muscularity etc..). Hence have removed ‘heritability’ from the title.

A section (lines 210-222) describing the statistical estimation of heritability has been added, but in short may be summarised in the output from our stats program below (e.g. for peak stride length):

REML variance components analysis

Response variate:     Peak_Stride_Length

Fixed model: Constant + AGEinYrs + TypeHORSE

Random model:          f_Horse_ID

Number of units:        406 (15 units excluded due to zero weights or missing values)

Residual term has been added to model

Sparse algorithm with AI optimisation

All covariates centred

Covariance structures defined for random model

 Covariance structures defined within terms:

Term   Factor Model  Order  No. rows

f_Horse_ID     f_Horse_ID     Fixed matrix ainv (inverse) 1            2690

Estimated parameters for covariance models

Random term(s)         Factor Model(order) Parameter      Estimate           s.e.

f_Horse_ID     f_Horse_ID     Fixed matrix   Scalar 0.2324             0.2289

Note: the covariance matrix for each term is calculated as G or R where

var(y) = Sigma2( ZGZ'+R ), i.e.      relative to the residual variance, Sigma2.

Residual variance model

Term   Model(order) Parameter       Estimate           s.e.

Residual           Identity              Sigma2             0.0470               0.00887

Tests for fixed effects

Sequentially adding terms to fixed model

Fixed term      Wald statistic                n.d.f.    F statistic         d.d.f.    F pr

AGEinYrs        9.94                                   1            9.94                    400.0   0.002

TypeHORSE  7.81                                   4            1.95                    393.7   0.101

Dropping individual terms from full fixed model

Fixed term      Wald statistic                n.d.f.    F statistic         d.d.f.    F pr

AGEinYrs        1.20                                   1            1.20     400.0   0.275

TypeHORSE  7.81                                   4            1.95     393.7   0.101

3941  DELETE [REDEFINE=yes] _vest,_h2,_nval,_tterm

3942  VKEEP [VESTIMATE=_vest]

3943  CALCULATE _nval = NVAL(_vest) - 1

3944  VARIATE [NVALUES=1] _h2

3945  CALCULATE _h2$[1] = _vest$[1]/(1+SUM(_vest$[!(1...#_nval)]))

3946  TXCONSTRUCT [TEXT=_tterm; METHOD=append] !t('f_Horse_ID')

3947  CAPTION 'Narrow-sense Heritability - h2'; STYLE=major

Narrow-sense Heritability - h2

3948  PRINT _tterm,_h2; HEADING='Term','h2'

Term   h2

f_Horse_ID     0.1886

All statistical methodology developed in results (below each table or figure) must be thoroughly and neatly explained in a specific section within material and methods and eliminated from results

I find it pretty standard to include the stats below a table or figure in any publication but have moderated what we have put there but not excluded entirely as I find it informative for the table/figure to be read alone. The statistical tests used are in the Statistics section of the Methods already, and so where appropriate have removed the extra words.

Results

L213 n = 35 of 3268 Delete “n=”.

Done. (line 234)

L218-221 The number of participants in the races is constant? This should be detailed in material and methods.

We do not have this information for all races, but do have each individual horses position in the race at 400m and 800m and in none of these does the position exceed 18. Therefore, we can say with some certainly that the races, or at least positional information is limited to 18.

Table 2 and 3 delete “Training type… “and “type horse…”

Done.

Explain better ‘spell or detraining’

Detailed (lines 371-374)

Explain ‘jump-out’ before

Jump-out explained previously at lines 166-169.

The section Training data, category of racehorse and predicting race outcome is confusing. Really the bibliography discussed is difficult to spin with your result. I don't know if that section should be improved or deleted.

The section 3.3 ‘Training data, type of racehorse and predicting race outcome’ is actually rather novel and would be a shame to delete upon the reviewers suggestion. It is rare for such different databases to come together as we have done here, and this paragraph describes whether any phenotypic data in training were able to predict race outcome. We only report the outcome (I.e. no huge effect of stride length). We think this will be of interest to the equine sports performance community. Our discussion aims to provide context to the study and a better understanding of the racing industry for the equine readers/scientists/greater public.

The bibliography (I.e. reference list) reflects the general lack of any research in this area.

Colts age range should be explained in material and methods

Yes, we agree. Ages for each biological sex have been added (lines 146-149)

Reviewer 2 Report

The paper is long and complicated. Clarifications are suggested

What is the main question? The main question addressed by the research was whether stride characteristics   explained success at racing in different types of races
The topic is relevant in the field because it could be used to increase a horse's racing success. 
It does add to the subject area compared with other published
material?
The conclusions are consistent with the evidence and arguments
presented, and address the main question posed?
the references are appropriate?

 Should make clear in summary and abstract that data were obtained before these horses raced and that measurement was compared to their later racing success.

 Were that examples of horses that  had shorter stridee, but were raced in longer races. Presumably that is buried in the tables, but should be explicitly stated. What are these groups? What determines quality?  For an international audience unfamiliar with racing  terms what is Listed ? 

127 n=65 of 3269  is the larger number the total number of horses in each class of race?

 It is not clear how  trainers are to select. Do they measure stride length and then  race the horse  in the type of race his stride length predicts he  will be most successful?

 342 Shouldn't this be in abstract? "racehorses with relatively short or long stride within sprint or  stayer categories, respectively did not predict race performance "

Fig 2 Any explanation for large changes at  training session 4 

 Minor problems. The punctuation and grammar is strange in 3rd sentence 

220

Fig 3  What is long and medium  and Group 

Should be repeated on non-turf horses 

Author Response

The paper is long and complicated. Clarifications are suggested.
What is the main question? The main question addressed by the research was whether stride characteristics explained success at racing in different types of races. The topic is relevant in the field because it could be used to increase a horse's racing success. It does add to the subject area compared with other published material? The conclusions are consistent with the evidence and arguments presented, and address the main question posed? the references are appropriate?
We thank the anonymous reviewer for the positive comments and have, where we agree, made changes to the manuscript as detailed below.
Should make clear in summary and abstract that data were obtained before these horses raced and that measurement was compared to their later racing success.
Yes, we agree. We have added to the simple Summary, Line 16-20, and it was previously mentioned in the abstract, (lines 28-29). 
Were that examples of horses that had shorter stride, but were raced in longer races. 
Presumably that is buried in the tables but should be explicitly stated. What are these groups? 
There were certainly examples of ‘pure sprinters’ and ‘pure stayers’ i.e. horses that only competed in races corresponding to the said distance. We categorized our horses accordingly as described previously in the manuscript (lines 142-147). There was variation in stride within each category (e.g. some stayers had relatively shorter stride than other stayers) as would be expected, but we have to stick to our ultimate categorization technique – based on the hard outcome of race performance in races for sprinters or stayers. Pure sprinters with short stride did not race in any other longer races. Pure stayers with long stride did not race any shorter distance races
What determines quality? 
‘Race quality’ modified to ‘race class’ (lines 133-139)
For an international audience unfamiliar with racing terms what is Listed?
Fine, now detailed in lines (lines 133-139)
127 n=65 of 3269 is the larger number the total number of horses in each class of race?
No, n=65 of 3269 races, as specified in text.
It is not clear how trainers are to select. Do they measure stride length and then race the horse in the type of race his stride length predicts he will be most successful?
Yes, to an extent that is one of our suggestions after putting this data together – of course, the first time to do that. We might expect such an approach to be a part of range of measures that determine what races the horse should be placed in. To select suitable race distance for any horse, trainers should consider aspects of pedigree information (if parents were sprinters/milers or stayers), objective stride data (high peak stride frequency or length, speed), experience (rider’s feeling, visual observation of the horse exercising on the track) and preferred track (hard vs soft ground). Then based on consecutive race results for that individual horse, reevaluate training/race distance if necessary.
342 Shouldn't this be in abstract? "racehorses with relatively short or long stride within sprint or stayer categories, respectively did not predict race performance "
In the abstract, line 32, we have "Relatively short or longer stride did not predict race success, but stayers had greater race success than sprinters (P<.001)."
Fig 2 Any explanation for large changes at training session 4
We have no obvious answer, other than using REML to predict the marginal means that is what we see (without any obvious outliers). We think that the anomaly is only in the stayers, of which we had relatively lower numbers, but cannot be excluded due to lack of outliers and in Fig 2b,c appears to some extent to follow the natural change from the previous sessions. We can add 
a note to say that some due caution can be used with such data (we were only looking for significant trends over time, there were none)
Minor problems. The punctuation and grammar is strange in 3rd sentence 220.
Addressed.
Fig 3 What is long and medium and Group
For the logistic regression we have to specifiy a referent category and so those are listed on the figure. We have hopefully made clearer.
Stride length: long, medium and short (categorized within the largest group, sprinters)
Race class: i.e. Group 1, Group 2, Group 3 and Listed.
Should be repeated on non-turf horses.
Yes, indeed that would be of interest, although racehorses in Australia do nearly all their fast work on turf and given that races are by definition at their fastest pace, then these datasets were appropriate for this analysis. Repeating in a different country that mostly run on dirt or sand would be of interest, but as yet not within our means. Considering the objectives of this study, speeds and track needed to reflect racing conditions as much as possible

Round 2

Reviewer 1 Report

  The authors have improved the manuscript in a suitable way for publication